# Hybrid Immunity from Gam-COVID-Vac Vaccination and Natural SARS-CoV-2 Infection Confers Broader Neutralizing Activity against Omicron Lineage VOCs Than Revaccination or Reinfection

**DOI:** 10.3390/vaccines12010055

**Published:** 2024-01-06

**Authors:** Sergey V. Kulemzin, Sergey V. Guselnikov, Boris G. Nekrasov, Svetlana V. Molodykh, Irina N. Kuvshinova, Svetlana V. Murasheva, Tatyana N. Belovezhets, Andrey A. Gorchakov, Anton N. Chikaev, Nikolai A. Chikaev, Olga Y. Volkova, Anna A. Yurina, Alexander M. Najakshin, Alexander V. Taranin

**Affiliations:** 1Institute of Molecular and Cellular Biology SB RAS, Novosibirsk 630090, Russia; skulemzin@mcb.nsc.ru (S.V.K.); sguselnikov@mcb.nsc.ru (S.V.G.);; 2AO Vector-Best, Novosibirsk 630090, Russia

**Keywords:** antibody, hybrid immunity, variants of concern, virus escape, neutralization breadth, COVID-19

## Abstract

SARS-CoV-2 has a relatively high mutation rate, with the frequent emergence of new variants of concern (VOCs). Each subsequent variant is more difficult to neutralize by the sera of vaccinated individuals and convalescents. Some decrease in neutralizing activity against new SARS-CoV-2 variants has also been observed in patients vaccinated with Gam-COVID-Vac. In the present study, we analyzed the interplay between the history of a patient’s repeated exposure to SARS-CoV-2 antigens and the breadth of neutralization activity. Our study includes four cohorts of patients: Gam-COVID-Vac booster vaccinated individuals (revaccinated, RV), twice-infected unvaccinated individuals (reinfected, RI), breakthrough infected (BI), and vaccinated convalescents (VC). We assessed S-protein-specific antibody levels and the ability of sera to neutralize lentiviral particles pseudotyped with Spike protein from the original Wuhan variant, as well as the Omicron variants BA.1 and BA.4/5. Individuals with hybrid immunity (BI and VC cohorts) exhibited significantly higher levels of virus-binding IgG and enhanced breadth of virus-neutralizing activity compared to individuals from either the revaccination or reinfection (RV and RI) cohorts. These findings suggest that a combination of infection and vaccination, regardless of the sequence, results in significantly higher levels of S-protein-specific IgG antibodies and the enhanced neutralization of SARS-CoV-2 variants, thereby underscoring the importance of hybrid immunity in the context of emerging viral variants.

## 1. Introduction

One of the salient features of SARS-CoV-2 is its rapid evolution. New VOCs emerge frequently and some of these new VOCs carry dozens of new mutations compared to the ancestral Wuhan (Wu-1) strain [1]. Despite a decrease in lethality, novel SARS-CoV-2 representatives of the Omicron lineage exhibit an increased transmissibility that contributes to the worldwide presence of the virus. Since these recent viral variants have emerged in populations with a complex history of immunizations and pre-established immunity, it is perhaps not surprising that they possess increased resistance to neutralization by the sera induced by previous exposure to the virus. For instance, the sera of vaccinated individuals and convalescent patients neutralized the Omicron variants BA.1, BA.2, and BA.4/5 that spread globally in 2022 significantly less effectively than the Wu-1 variant [2,3,4]. The activity against the BQ.1.1, XBB, and XBB.1.5 variants, which became prevalent at the end of 2022 and the beginning of 2023, has been reported to have reduced by almost two orders of magnitude. Moreover, none of the recent viral variants can be neutralized by the antiviral monoclonal antibodies approved for limited therapeutic use [5,6].

We previously reported the isolation of a panel of human monoclonal antibodies displaying highly potent neutralization of the Wu-1 variant of the virus [7]. One antibody from this panel showed high activity against early omicron variants [8], but none of them neutralized BQ.1.1, XBB, or XBB.1.5. Therefore, a deeper understanding of how broad immunity may be formed would be valuable for improving vaccination strategies and guiding identification of next-generation, pan-sarbecovirus therapeutic antibodies. The existing data indicate that the breadth of neutralization is largely associated with repeated antigenic stimuli. Numerous studies have reported that RNA vaccine booster shots contribute to the development of antibodies capable of neutralizing new virus variants, especially when the booster is heterologous [9,10,11,12,13,14]. Increased breadth of neutralization has also been observed in the case of so-called hybrid immunity, induced as a result of either infection breakthrough after vaccination or the vaccination of convalescent individuals [11,15,16].

In the present study, we aimed to investigate whether there are differences in the breadth of neutralization in individuals whose immune system was repeatedly stimulated with either natural SARS-CoV-2 antigens or antigens from the GAM-COVID-Vac (Sputnik V) vaccine. This vaccine belongs to the first generation of COVID-19 vaccines and is based on the use of the “wild-type” Spike protein of SARS-CoV-2 encoded by two replication-incompetent adenovirus vectors (rAd26 and rAd5) [17]. Previously, it was found that GAM-COVID-Vac induced a broader immune response in convalescents compared to naive recipients [18,19]. In the present study, we set out to examine individuals from four groups of donors: those who were booster vaccinated, those who were twice infected but never vaccinated, those who were infected after vaccination, and those who were vaccinated following SARS-CoV-2 infection. An important aim of our study was to identify the donors with pronounced virus-neutralizing activity that was similar for the Wu-1 and Omicron variants. We hypothesized that such characteristics would be consistent with the presence of broadly neutralizing antibodies. We assessed the titers of antiviral antibodies and their neutralizing activity against lentiviruses pseudotyped with S-proteins from three SARS-CoV-2 variants: the original Wu-1 strain as well as the Omicron variants BA.1 and BA.4/5. It is noteworthy that the latter two variants emerged after the blood samples were collected, i.e., the donors were never exposed to BA.1 or BA.4/5. The results obtained show that antibody titers and breadth of neutralization are the most pronounced in individuals with hybrid immunity. In addition, our data indicate that vaccination with GAM-COVID-Vac following infection enhances the patient’s ability to neutralize new mutant VOCs.

## 2. Materials and Methods

### 2.1. Cell Lines

The HEK293T cell line was obtained from ATCC (CRL-3216). The HEK293T-hACE2 cell line has been described previously [7]. The cell lines were cultured in the IMDM medium (#12440053, Thermo Fisher Scientific, Waltham, MA, USA) supplemented with 10% FBS (FBS-12B, Capricorn, Ebsdorfergrund, Germany), 100 U/mL penicillin, and 100 μg/mL streptomycin (#15140122, Thermo Fisher Scientific). Cells were passaged twice a week, or the day before transfection or transduction.

### 2.2. Serum Collection and Ethics Approval

This study complies with the Declaration of Helsinki (1975) and was approved by the Ethics Committee on Animal and Human Research of the Institute of Molecular and Cellular Biology (Novosibirsk, Russia), No. 02/21 from 4/8/2021. All the procedures for obtaining written informed consent were also approved by the Ethics Committee. Two copies of the consent were signed, one copy was provided to the participants and one copy to the Institute of Molecular and Cellular Biology (Novosibirsk, Russia). Serum samples were obtained from otherwise healthy donors who were 18–70 years old of both sexes. Patient cohorts were not balanced by age and gender.

### 2.3. ELISA-Based Detection of IgG, IgM, and IgA Antibodies Specific to SARS-CoV-2 Proteins

To measure the concentration of full-length trimerized S protein-specific IgG antibodies in patients’ sera, we used ELISA kit D-5505 (Vektor-Best, Novosibirsk, Russia).

The method is an indirect solid-phase enzyme immunoassay. In the first stage of the analysis, specific antibodies (IgG) that are present in the test samples bind to the surface-immobilized recombinant antigen of SARS-CoV-2 on the wells of the plate (full-length trimerized surface glycoprotein S, including the receptor-binding domain (RBD)). In the second stage, conjugates of monoclonal antibodies to human IgG with horseradish peroxidase interact with the “antigen–antibody” complexes. Upon incubation with a TMB substrate, the solution in the wells containing specific antibodies undergoes coloration. The optical density of the solution in each well is proportional to the concentration of antibodies to the SARS-CoV-2 protein in the analyzed sample. Optical density was measured by the Thermo Scientific Multiskan Microplate Reader.

This kit was calibrated according to the WHO standard and is approved for quantitative clinical use. Full-length trimerized S protein-specific IgA and (RBD+N)-specific IgM were measured in a semi-quantitative way using D-5503 and D-5502 kits, respectively (Vektor-Best, Novosibirsk, Russia).

### 2.4. Plasmids

Nluc-encoding plasmid pCHD-Nluc was generated by cloning Nluc cDNA (Uniprot #Q9GV45) into pCDH-CMV-MCS-EF1α-copGFP vector using XbaI and BamHI sites.

To generate S-protein encoding plasmids, cDNAs for Wu, BA.1, and BA4/5 Spike proteins devoid of 19 C-terminal amino acid residues were obtained by gene synthesis (Azenta) and cloned to pCAGGS using XbaI and NotI sites. Plasmid DNA was purified using the Qiagen Endofree plasmid maxi kit, and the identity of the plasmids was verified by Sanger sequencing. After purification, plasmid DNA was precipitated using ethanol precipitation and reconstituted in sterile DNAse free water using the aseptic technique.

### 2.5. Production of S-Pseudotyped Lentiviral Particles

Production of S-pseudotyped lentiviral particles has been described previously [7]. Briefly, HEK293T cells were transfected with a mixture of plasmids psPAX2, pCDH-Nluc, and a pCAGGS-SpikeΔ19 plasmid encoding either the Wu-1 or VOCs SARS-CoV-2 S protein lacking 19 C-terminal amino acid residues. Lentiviral particles were purified by centrifugation from supernatants from the conditioned medium 48 h after transfection. Preparations were titrated on HEK293T-Ace2 cells to determine the number of functional viral particles. Because of the extremely high levels of Nluc-driven luminescence, only 1000 viral particles per 15,000 cells in one well of a 96-well plate were used in the neutralization experiments.

### 2.6. SARS-CoV-2 S-Pseudotyped Lentivirus Neutralization Assay

HEK293T-hACE2 cells stably expressing human ACE2 were seeded at a density of 15,000 cells/well in a 96-well plate on the day before the neutralization assay. Heat-inactivated sera were serially diluted in Opti-MEM supplemented with 2.5% heat-inactivated FBS. We used twofold dilutions of serum samples in a range from 1:4 to 1:1024–65536 (depending on the serum’s neutralization potency). Serum dilutions were co-incubated with S-pseudotyped lentiviral particles for 30 mins at 37 °C in a volume of 100 µL. After preincubation, an antibody/S-pseudotyped lentivirus mixture was added to the HEK293T-hACE2 cells and the plate was placed in a CO_2_ incubator. Then, 48 h following transduction, the cells were washed with PBS and lysed in PBS+0.2% Triton X-100. Luminescence intensity was measured by Luminoscan (Thermo Fisher Scientific) 100 ms after the addition of the substrate to the well (1.25 µg of freshly prepared h-coelenterazine in 50 µL of PBS) over a period of 3 s. Integral fluorescence was used for the additional calculations. The half-maximal inhibitory dilution (ID50) was determined by non-linear regression as the concentration of antibody dilution neutralized 50% of the pseudotyped lentivirus. Data from two independent experiments were used.

### 2.7. Statistics

Statistical analysis of the significance of differences between groups was performed using the Kruskal-Wallis one-way analysis of variance. Spearman’s correlation analysis was conducted to calculate the correlation.

## 3. Results

In the present study, we examined 65 serum samples that were collected from four cohorts of donors characterized by repeated stimulation with SARS-CoV-2 antigens. The samples were collected in the second half of 2021, long before the Omicron lineage was prevalent in Russia [20,21]. During the years 2020–2022, all of the individuals in this study were PCR-tested for SARS-CoV-2 infection on a regular basis, at least twice per week, as the Vector-Best employees in the course of the company’s anti-COVID-19 program. The group of fully vaccinated individuals (revaccinated, RV) included 15 serum samples from individuals who had received a booster dose of Gam-COVID-Vac (“Sputnik V”), had no self-reported symptoms of SARS-CoV-2 infection, and were negative for anti-nucleocapsid antibodies. The interval between vaccinations in this group ranged from 102 to 225 days (Appendix A). The time that elapsed between the second vaccination and blood collection ranged from 21 to 105 days.

The RI (reinfected) group included 13 samples from individuals who had been infected twice during the pandemic but were never vaccinated. Antibodies against the nucleocapsid antigen were detected in all the samples. The specific variant of coronavirus responsible for the infection was not determined. However, eight individuals in this group were first infected in 2020 during the spread of the Wuhan variant and were reinfected in the second half of 2021, primarily during the circulation of the Delta variant. Two individuals in this group were infected twice in 2021, with intervals between infections of 3 and 4 months, respectively. There was no PCR data for secondary infections for three individuals. Nevertheless, they were included in this group based on the description of typical infection symptoms and an increase in the titers of antiviral antibodies relative to the levels documented after the first infection. One individual from this group had severe COVID-19 during the initial infection. All the others developed mild-to-moderate illness. The second infection caused mild-to-moderate illness in eight individuals and was asymptomatic in five (Appendix A). The period between the secondary infection and blood collection ranged from 30 to 280 days.

The BI (breakthrough infected) group included 23 samples from individuals who had received a single course of Gam-COVID-Vac vaccination and subsequently became infected 23 to 235 days later. Blood samples from this cohort were collected 21 to 150 days after a PCR-confirmed diagnosis. Finally, the VC (vaccinated convalescents) group included 14 samples from individuals who were vaccinated after recovering from the infection. The period between recovery and vaccination varied from 121 to 420 days, and the period between vaccination and blood collection ranged from 35 to 150 days. In the present study, we did not take into account gender, the average age of donors, or the severity of the infection they had experienced.

### 3.1. S-protein-Specific Antibody Levels

We assessed the quantity of full-length trimerized S protein-specific IgG antibodies in the examined samples to determine binding antibody units per ml (BAU/mL), calibrated according to the WHO standard (the First WHO International Standard for anti-SARS-CoV-2 immunoglobulin (human), NIBSC code: 20/136. BAU). The results showed a significant difference between patients with hybrid immunity (BI and VC) and those who were either only vaccinated or only infected (RV and RI) (Figure 1, left panel). The median values were as follows: for the VC group-2038 (IQR: 803-4468), for the BI group-2123 (IQR: 969-4695), for the RI group-441 (IQR: 188-732), and for the RV group-264 BAU/mL (IQR: 86-1022) (Figure 1 (left panel), Appendix A). The difference between the BI and VC as well as between the RV and RI groups was not significant. Therefore, the combination of infection and vaccination, regardless of the sequence in which they occurred, led to significantly higher levels of S-protein-specific IgG antibodies than reinfection or revaccination alone.

We did not find a significant difference between the four groups in terms of (RBD+N)-specific IgM antibody levels (Appendix A). However, the concentration of S protein-specific IgA antibodies was significantly higher in the RI, BI, and VC groups compared to the RV group. This is concordant with the known fact of the rapid drop of mRNA vaccine-induced serum IgA compared to its kinetics in naturally infected or convalescent vaccinated individuals [22].

### 3.2. Neutralization of Pseudoviruses

To evaluate the ability of the serum samples to neutralize SARS-CoV-2, we used lentiviral particles pseudotyped with three different Spike protein variants, namely the original Wuhan variant (Wu-1), as well as two Omicron lineage variants: BA.1 and BA.4/5, with the Spikes of BA.4 and BA.5 being identical (Figure 2). All the samples contained antibodies capable of neutralizing the Wu1 variant. However, as with the analysis of the antiviral antibody levels, the serum samples from the VC and BI groups demonstrated significantly higher pseudovirus-neutralizing activity compared to the RV and RI groups. The GMT ID50 values for the BI and VC groups were 2628 (IQR: 875–8165) and 2507 (IQR: 1333–4914), respectively. In the RV and RI groups, the GMT ID50 values were 197 (IQR: 50–864) and 369 (IQR: 134–914), respectively (Figure 2A, Appendix A).

Unlike with Wu-1 Spike-pseudotyped lentiviruses, 7 out of 15 samples in the RV group displayed no neutralizing activity against the lentiviral particles pseudotyped with BA.1 Spike. GMT ID50 was 33 (IQR: 10–84). Two samples with activity below the detection threshold were found in the RI group and one in the BI group. The GMT ID50 in these groups was 36 (IQR: 20–95) and 177 (IQR: 53–390), respectively. In the VC group, neutralizing activity against BA.1 was detected in all the samples with a GMT ID50 of 402 (IQR: 175–876) (Figure 2A, Appendix A). The difference was highly statistically significant between groups with hybrid immunity (BI and VC) and either the infection or vaccination-only groups (RV and RI).

Similar results were obtained when analyzing neutralizing activity against the BA.4/5 variant. In 8 out of the 15 samples in the RV group, the level of neutralization was below the detection threshold. No activity was detected in 1 out of 13 samples in the RI group and 3 out of 23 samples in the BI group. The geometric mean ID50 titers of neutralizing antibodies were 18 (IQR: 10–38), 41 (IQR: 18–94), 182 (IQR: 33–671), and 223 (IQR: 95–617) in the RV, RI, BI, and VC groups, respectively (Figure 2A, Appendix A). In this case, the difference was again highly significant between the hybrid immunity groups (BI and VC) and the RV samples.

The drop in neutralizing activity in the studied serum samples against the BA.1 and BA.4/5 variants relative to the Wuhan variant is shown in Figure 2B. In the case of BA.1, the fold change in GMT ID50 values ranged from 6 to 15 in different groups. In the case of BA.4/5, the reduction in neutralization ranged from 9 to 14 fold. However, in some individuals, we found relatively small differences in neutralizing activity titers against the three virus variants studied. For example, the neutralization activity of samples #59 (BI group) and 63, 67, 69 (VC group) against the Wu-1 variant was higher than the activity against BA.1 and BA.4/5 by a factor ranging from 1.3 to 2.9 (Figure 2B).

It is still unclear how applicable the quantity of antibodies specific to the S-protein of the Wu-1 strain is for assessing the level of virus-neutralizing activity against novel VOCs. To answer this question, we assessed the correlation of the detected quantitative indicators using Spearman’s rank correlation. Overall, when analyzing the entire pool of samples, the level of neutralizing activity against the Wu-1, BA.1, and BA.4/5 variants did not correlate with IgM or IgA class antibodies, but showed a highly significant correlation with IgG antibodies concentration (*p* < 0.0001 for all of the variants; r = 0.81, r = 0.82, and r = 0.78, respectively) (Figure 3). A significant correlation was observed in the entire sample when comparing the neutralizing activity against different virus variants (Figure 3). However, when analyzing separate groups of study participants, this pattern was not always reproduced. For instance, in the VC group, the neutralizing activity against both BA.1 and BA.4/5 correlated with each other and with the level of IgG antibodies, but none of these parameters correlated with activity against Wu-1. In the RI group, weak or no correlation was observed between the parameters studied.

## 4. Discussion

Currently, the majority of the population has some degree of immunity to SARS-CoV-2, either as a result of infection or vaccination. Many individuals have received booster shots. Many vaccinated individuals have also been infected with the virus. In the present study, we investigated how the repeated stimulation of the immune system through natural infection and/or Gam-COVID-Vac vaccination affects the ability of humoral immunity to neutralize novel variants of the SARS-CoV-2 virus and the emergence of new variants. To achieve this, we compared the levels of virus-specific antibodies and the neutralizing activity against three SARS-CoV-2 variants in four groups of donors with differing viral antigen restimulation contexts: revaccinated, twice infected unvaccinated, naturally infected after vaccination, and vaccinated after recovery from a SARS-CoV-2 infection.

All of the samples were collected before the emergence of the Omicron variants in Russia [20,21]. In the RV and VC groups, viral antigens were associated with the earliest SARS-CoV-2 Wu-1 variant in both the first and second encounters. The Gam-COVID-Vac vaccine was developed in 2020, based on the non-stabilized S protein of the Wuhan variant [17]. In the RI and BI groups, during the second encounter, the antigen could have been derived from early variants, or the Delta variant, which was circulating in Russia in the second half of 2021 [20,21]. We investigated whether there were differences between the RV, RI, BI, and VC groups in terms of their ability to resist new virus variants, BA.1 and BA.4/5, belonging to the Omicron lineage. BA.1, which was the first representative of this lineage, emerged at the end of 2021. The BA.5 variant spread in the second half of 2022. Both variants are known to exhibit increased resistance to the immune response compared to the Wuhan variant of the virus and, among other features, are not neutralized by most previously approved SARS-CoV-2-specific monoclonal therapeutic antibodies [4,23].

Our results show a significant difference between the hybrid immunity (BI and VC) groups compared to the RV and RI groups, in terms of the levels of antiviral antibodies and the ability to neutralize one or both variants from the Omicron lineage. We see that vaccination with Gam-COVID-Vac after infection significantly increases antibody titers capable of neutralizing the Omicron lineage variants compared to the booster vaccination or double infection. While infection should be avoided because of the risk of complications, vaccination may be beneficial for recovered patients.

Interestingly, we show here that hybrid immunity is more effective than immunity induced by two natural infections. These results contradict some other data that suggest the advantages of the immune response induced by natural infection compared to the immune response to vaccine immunogens [18,24,25,26]. There may be several explanations. The first is that the low level of antigen-binding and neutralizing antibodies in the RI group is a result of milder symptoms occurring during reinfection. Indeed, while the initial infection caused mild-to-moderate and severe disease in this group, reinfection was asymptomatic in five out of thirteen individuals. Secondly, it is known that infection provides sustained lung mucosal immunity, both cellular and humoral, unlike systemic immunization, which induces little to no mucosal antibodies in the secretions (lungs/saliva) [27]. The antibodies in the secretions from the initial infection may be a barrier to reinfection, decreasing the production of neutralizing antibodies. Third, it should be noted that some individuals may mount a weak immune response to SARS-CoV-2 infection [28]. It is possible that such individuals are more likely to experience reinfection and, therefore, we cannot exclude the possibility that at least some of the participants we selected were ‘non-responders’. Understanding the reasons behind relatively weak antiviral immunity in individuals with a history of two infections requires further investigation.

The observed correlation between neutralizing activity against different virus variants and the levels of antiviral IgG in peripheral blood indicates that higher antibody levels provide better protection against newer viral variants. This conclusion aligns with the data obtained by other authors [28,29]. However, this does not rule out the possibility of a lack of protection due to individual variations in immune response. Our data also suggest the possible influence of the context in which restimulation by viral antigens occurred. For example, no significant correlations were observed between the levels of neutralizing activities and antibodies in individuals who had been infected twice. From the perspective of new pan-sarbecovirus therapeutic agent discovery, individuals whose serum samples show relatively small differences in neutralizing activity against both Omicron variants, on the one hand, and the Wu-1 variant, on the other hand, are of particular interest. In our study, we found four participants from the BI and VC groups who showed a difference ranging from 1.5 to 2.9 fold, despite an approximately tenfold difference in the GMT values. These indicators suggest the presence of antibodies with the ability to neutralize a wide range of virus variants. We believe such individuals are of special interest as potential donors for the identification of novel broadly neutralizing monoclonal antibodies.

## Figures and Tables

**Figure 1 vaccines-12-00055-f001:**
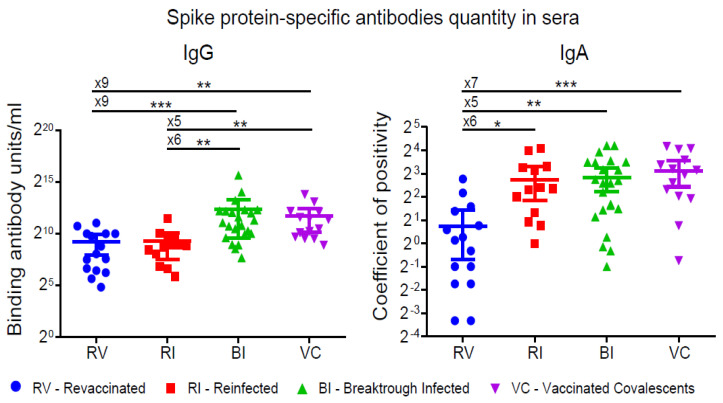
IgG and IgA virus-binding activity in sera from individuals after repeated stimulation of the immune system with either natural and/or Gam-COVID-Vac vaccine SARS-CoV-2 antigens. Wu-1 S-protein-specific IgG and IgA levels were measured by ELISA and shown as binding antibody units (IgG) or coefficient of positivity (IgA). Comparisons show the fold change in median values (x5–x9). Statistical significance was determined using the Kruskal–Wallis test (* *p* < 0.1, ** *p* < 0.01, *** *p* < 0.001.

**Figure 2 vaccines-12-00055-f002:**
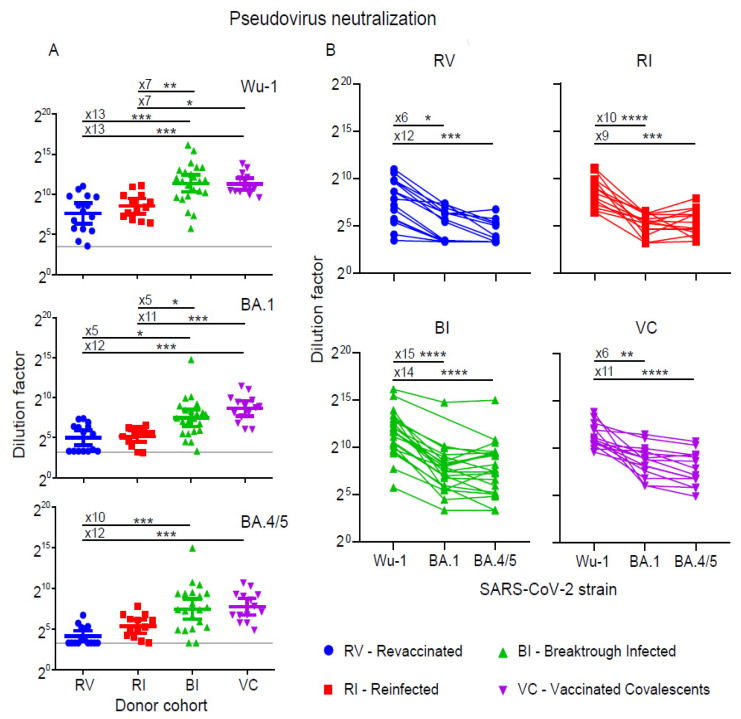
Neutralizing activity against Wu-1, Omicron BA.1, and Omicron BA.4/5 variants of pseudoviruses in sera from revaccinated (RV), reinfected (RI), breakthrough infected (BI), and convalescent vaccinated (CV) groups. (**A**) Serum neutralizing ID50 titers. Comparisons show the fold change in median values (x5–x15). Grey line shows the detection threshold. (**B**) Individual comparison of neutralizing ID50 titers against the variant pseudoviruses in the indicated groups. The fold changes in geometric mean ID50 titer (GMT) are shown. Statistical significance was determined using the Kruskal–Wallis test (* *p* < 0.1, ** *p* < 0.01, *** *p* < 0.001, **** *p* < 0.0001).

**Figure 3 vaccines-12-00055-f003:**
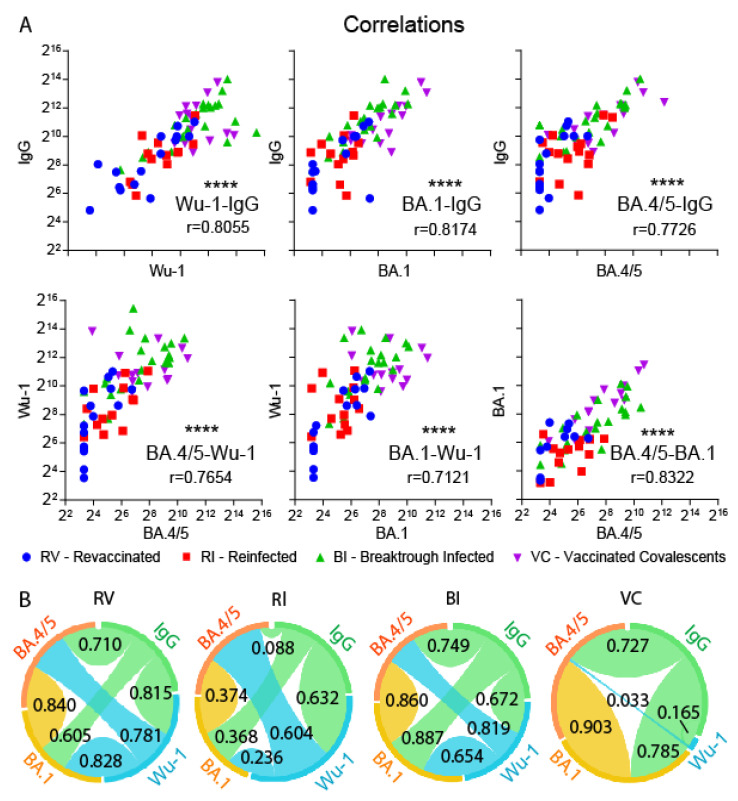
Spearman’s correlation between Wu-1 S-protein binding (IgG) and neutralizing activity against Wu-1, BA.1, and BA.4/5 variants (**A**, upper row), or between neutralizing activity against Wu-1, BA.1, and BA.4/5 variants (**A**, lower row) in sera from individuals representing revaccinated (RV), reinfected (RI), breakthrough infected (BI), and vaccinated convalescent (VC) groups. Chord diagrams (**B**) show Spearman’s correlations between virus neutralization values (ID50) of Wu-1, BA.1, and BA.4/5 variants and the level of Wu-1 S-protein binding IgG in different cohorts. Chord widths are proportional to the correlation coefficients indicated on the corresponding chords. **** indicates *p* < 0.0001.

## Data Availability

Data supporting reported results can be requested from the corresponding author.

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
