# Peer review of "Hybrid Immunity from Gam-COVID-Vac Vaccination and Natural SARS-CoV-2 Infection Confers Broader Neutralizing Activity against Omicron Lineage VOCs Than Revaccination or Reinfection"

_vaccines, 2024, doi:10.3390/vaccines12010055_

Round 1
Reviewer 1 Report
Comments and Suggestions for Authors
The Manuscript examines the quality of the humoral immune response to the COVID 19 spike protein after vaccination/infection or combinations thereof. It then looks at the neutralizing ability of these antibodies.
The data is well presented and easy to understand.
Comments:
1. It is obvious that people that have been infected and not ever vaccinated would have different immunity to the vaccinated only individuals. It is interesting that people that have had two rounds of COVID infection are not the most protected in this study. Do you believe that COVID antibodies in the lungs from the initial infection may prevent/dampen a subsequent infection of COVID 19 preventing the production of high titre neutralising antibodies?
2. On your graphs there is an x followed by a number near the Astérix to indicate significance on the graph. Can you please define what this means in the figure legend. Also, the I assume that the grey line is the limit of the assay. Please include these details in the figure legends.
Author Response
- It is obvious that people that have been infected and not ever vaccinated would have different immunity to the vaccinated only individuals. It is interesting that people that have had two rounds of COVID infection are not the most protected in this study. Do you believe that COVID antibodies in the lungs from the initial infection may prevent/dampen a subsequent infection of COVID 19 preventing the production of high titreneutralising antibodies?
Answer: We thank the Reviewer for this quite probable explanation. It was added to the discussion section.
- On your graphs there is an x followed by a number near the Astérix to indicate significance on the graph. Can you please define what this means in the figure legend. Also, the I assume that the grey line is the limit of the assay. Please include these details in the figure legends.
Answer: Figure legends were changed accordingly.
Reviewer 2 Report
Comments and Suggestions for Authors
The authors analyzed the relationship between the history of patient's repeated exposure to SARS-CoV-2 antigens and the breadth of neutralization activity using four types of patients’ sera: ReVaccinated, RV; ReInfected, RI; breakthrough infected, BI; vaccinated convalescents, VC. The method adopted is unique and smart, and the results provided are clear. However, there are concerns that need to be addressed for publication.
1. The last paragraph of Introduction (lines 58–84) is inappropriate and redundant. Most of the topic dealt in this paragraph should be moved to Results and/or Discussion sections.
2. Throughout the text, indicate the details of Figures and Tables (including those of Supplementaries) more specifically, e.g. lines 210–211, with BA.1 Spike. GMT ID50 was 33 (IQR: 10-84) (Figure 2A, middle panel, Table Sx).
3. Lines 189–190.
Quote appropriate references that indicate the positive relationship between mucosal immune response (secretory dimeric IgA) and serum IgA (monomeric IgA).
4. Lines 184–185.
‘RBD specific IgG antibodies’ is not appropriate in this context and should be replaced with ‘neutralizing IgG antibodies.’ The authors just quantified anti-S IgG antibody (Ab), as described in “To measure the concentration of full-length trimerized S protein-specific IgG antibodies in patients’ sera” (lines 102–103). This Reviewer assume that the authors wrote this ‘RBD specific IgG antibodies’ based on the notion that an Ab that binds to RBD of S protein exhibits neutralizing activity. However, there is a possibility that Abs recognizing the adjacent region(s) of RBD occasionally exhibits neutralizing activity as demonstrated for influenza A virus.
5. Explain the details of Chord diagrams (Fig. 3) in Materials and Methods.
6. Line 179. Where is Fig. 1a?
Author Response
- The last paragraph of Introduction (lines 58–84) is inappropriate and redundant. Most of the
topic dealt in this paragraph should be moved to Results and/or Discussion sections.
Answer: Although it is a usual practice to describe the goals of the study and briefly summarize the conclusions in the Introduction section, we agree that some of the information appears to be redundant. To correct this, we removed the last three sentences in the indicated paragraph.
- Throughout the text, indicate the details of Figures and Tables (including those of Supplementaries) more specifically, e.g. lines 210–211, with BA.1 Spike. GMT ID50 was 33 (IQR:
10-84) (Figure 2A, middle panel, Table Sx).
Answer: The text was changed accordingly.
- Lines 189–190.
Quote appropriate references that indicate the positive relationship between mucosal immune
response (secretory dimeric IgA) and serum IgA (monomeric IgA).
Answer: Thank you for pointing this out, as our expression was indeed incorrect. The text has been revised accordingly.
- Lines 184–185.
‘RBD specific IgG antibodies’ is not appropriate in this context and should be replaced with
‘neutralizing IgG antibodies.’ The authors just quantified anti-S IgG antibody (Ab), as described in“To measure the concentration of full-length trimerized S protein-specific IgG antibodies in patients’ sera” (lines 102–103). This Reviewer assume that the authors wrote this ‘RBD specific IgG antibodies’ based on the notion that an Ab that binds to RBD of S protein exhibits neutralizing activity. However, there is a possibility that Abs recognizing the adjacent region(s) of RBD occasionally exhibits neutralizing activity as demonstrated for influenza A virus.
Answer: Corrected.
- Explain the details of Chord diagrams (Fig. 3) in Materials and Methods.
Answer: Figure legend was changed accordingly.
- Line 179. Where is Fig. 1a?
Answer: Corrected to (Fig.1 left panel)
Reviewer 3 Report
Comments and Suggestions for Authors
This is an interesting study pointing out a difference between the presence of anti-SARS-CoV neutralizing antibodies between patients who were vaccinated and infected (hybrid immunity), vs fully vaccinated or re-infected patients. A few points to be addressed by the authors:
(a) As far as I can understand the study is retrospective and not prospective. This creates a major drawback as to how SARS-CoV infection was assessed (I assume it was assessed through history or measuring IgM/IgG antibodies). It is widely accepted that SARS-CoV-2 cases with asymptomatic or mild infection do not develop high titers of antibodies in comparison with patients experiencing mild to severe infection. How the authors have documented this in their patient sample?
(b) The problem of hybrid immunity has been extensively addressed as well as the common practice or suggestion to combine doses of different SARS-CoV-2 vaccines (i.e. first doses Pfizer and after Moderna or Astra-Zeneca plus Moderna). However, it should be noted by the authors that the risk of re-infection is not an option to our patients, since the risk of complications such as long-COVID is increased with after re-infection. This should also be mentioned within the paper in their conclusions.
(c) It would be interesting to add a group of patients which chose to be vaccinated with different vaccines in comparison with the other groups and document whether there is a different response to the production of neutralizing antibodies.
Comments on the Quality of English LanguageMinor spelling is needed.
Author Response
This is an interesting study pointing out a difference between the presence of anti-SARS-CoV neutralizing antibodies between patients who were vaccinated and infected (hybrid immunity), vs fully vaccinated or re-infected patients. A few points to be addressed by the authors:
(a) As far as I can understand the study is retrospective and not prospective. This creates a major drawback as to how SARS-CoV infection was assessed (I assume it was assessed through history or measuring IgM/IgG antibodies). It is widely accepted that SARS-CoV-2 cases with asymptomatic or mild infection do not develop high titers of antibodies in comparison with patients experiencing mild to severe infection. How the authors have documented this in their patient sample?
Answer: In the Results section, we additionally mentioned that all the individuals studied here were PCR- tested for SARS-CoV-2 infection on a regular basis at least twice per week as the Vector-Best employees as part of the company’s anti-Covid19 program.The information about illness severity was added to the description of the RI group. Furthermore, the probable effect of asymptomatic infection on the levels of neutralizing antibodies was discussed.
(b) The problem of hybrid immunity has been extensively addressed as well as the common practice or suggestion to combine doses of different SARS-CoV-2 vaccines (i.e. first doses Pfizer and after Moderna or Astra-Zeneca plus Moderna). However, it should be noted by the authors that the risk of re-infection is not an option to our patients, since the risk of complications such as long-COVID is increased with after re-infection. This should also be mentioned within the paper in their conclusions.
Answer: The discussion section was corrected accordingly. Natural infection is definitely not an option and should be avoided.
(c) It would be interesting to add a group of patients which chose to be vaccinated with different vaccines in comparison with the other groups and document whether there is a different response to the production of neutralizing antibodies.
Answer: We are also quite curious about how combination of Gam-COVID-Vac with other vaccines (especially mRNA-based vaccines) would shape the immunity. Unfortunately, there was no wide usage of such vaccines in the Russian Federation, so assembling such a cohort would be extremely challenging.
Reviewer 4 Report
Comments and Suggestions for Authors
This is another a series of papers describing the humoral immune response to SARS-CoV-2 following infection and/or immunization. In this paper they compare effects of repeated antigen exposure on the breadth of antibody neutralization. The studies are well performed and presented, but do not cut any new ground. Larger numbers of patients in syudies such as these would be useful.
Author Response
This is another a series of papers describing the humoral immune response to SARS-CoV-2 following infection and/or immunization. In this paper they compare effects of repeated antigen exposure on the breadth of antibody neutralization. The studies are well performed and presented, but do not cut any new ground. Larger numbers of patients in studies such as these would be useful.
Answer: Thank you for your review. Hybrid immunity is indeed a well-studied phenomenon with multiple teams working in this field. The novelty of our research is that we studied hybrid immunity after Gam-COVD-Vac vaccination. Approximately 80 million people were vaccinated with Gam-COVID-Vac in Russia, and almost no other vaccines were used. Meanwhile, to our knowledge, hybrid immunity involving natural infection and Gam-COVID-Vac vaccination has received very little if any attention, so our research could fill this gap. Higher numbers of patients could definitely increase accuracy, but we hope that the statistical methods used in this research adequately support the conclusions based on the cohorts assembled for our study.
Reviewer 5 Report
Comments and Suggestions for Authors
In the current study, the authors investigated whether there were differences in the breadth of SARS-CoV-2-neutralizing activity against the Wu-1, and the new Omicron-derived variants, BA.1 and BA.4/5 in the sera of four group donors with differing viral antigen restimulation contexts: revaccinated (RV), twice infected not vaccinated (RI), breakthrough infected after vaccination (BI), and vaccinated after recovery from a SARS-CoV-2 infection (VC). The vaccine used here was the GAM-Covid-Vac (Sputnik V) that was the first generation of COVID-19 vaccines based on the use of the Wu-1 Spike protein encoded by replication-incompetent adenovirus vectors. They found that the individuals with hybrid immunity (BI and VC) exhibited significantly higher levels of virus-binding IgG and enhanced breadth of neutralizing activity compared to individuals with either revaccination or reinfection (RV and RI). These findings suggest that the importance of hybrid immunity should be underscored in the context of emerging viral variants. The manuscript is well-written and the findings are interesting. However, I have raised several points which need to be clarified. These are given below.
Specific points:
1) The authors showed here that hybrid immunity is more effective for variants than revaccination or reinfection. So, how should we take advantage of these findings against newly emerged virus variants?
2) Do the authors think that hybrid immunity is more effective even when using mRNA vaccines instead of GAM-Covid-Vac?
3) I suppose that reactivation with GAM-Covid-Vac induced the production of adenovirus-specific antibodies. Is this one of the reasons why the individuals with RV produced lower levels of anti- SARS-CoV-2 antibodies?
4) Do the authors consider that hybrid immunity is more effective in the induction of cellular immunity as well?
Author Response
Specific points:
1) The authors showed here that hybrid immunity is more effective for variants than revaccination
or reinfection. So, how should we take advantage of these findings against newly emerged virus variants?
Answer: In discussion, we corrected the text to emphasize that vaccination after recovery may be beneficial.
2) Do the authors think that hybrid immunity is more effective even when using mRNA vaccines instead of GAM-Covid-Vac?
Answer: Yes, the efficiency of hybrid immunity in the case of mRNA vaccines has been described in numerous studies [For instance see refs 11,15,16]
3) I suppose that reactivation with GAM-Covid-Vac induced the production of adenovirus-specificantibodies. Is this one of the reasons why the individuals with RV produced lower levels of anti-SARS-CoV-2 antibodies?
Answer: In a recently published paper Byazrova et al have shown that Anti-ADV immunity does not compromise SARS-COV-2 neutralizing antibody responses following Gam-COVID-Vac booster vaccination. [Byazrova MG, Astakhova EA, Minnegalieva AR, Sukhova MM, Mikhailov AA, Prilipov AG, Gorchakov AA, Filatov AV. Anti-Ad26 humoral immunity does not compromise SARS-COV-2 neutralizing antibody responses following Gam-COVID-Vac booster vaccination. NPJ Vaccines. 2022 Nov 15;7(1):145. doi: 10.1038/s41541-022-00566-x.
4) Do the authors consider that hybrid immunity is more effective in the induction of cellular immunity as well?
Answer: We didn’t discuss the cellular immunity as we did not study it. There is evidence that individuals with hybrid immunity show better correlated adaptive immune responses compared to those only vaccinated [doi: 10.1038/s41467-023-36250-4.]
Round 2
Reviewer 2 Report
Comments and Suggestions for Authors
The authors have responded appropriately to my concerns. I think it is acceptable.
Author Response
We are thankful to the reviewer for his/her comment.
Reviewer 3 Report
Comments and Suggestions for Authors
All my comments have been adequately addressed.
Comments on the Quality of English LanguageEnglish needs minor editing
Author Response
We are thankful to the reviewer for his/her recommendation to improve English in our manuscript. Indeed, given that none of the authors are native English speakers, the manuscript was sent for editing by a professional proofer prior to submission to the journal. We strive for better presentation of our results and would be very grateful to the Reviewer for pointing out the sentences/expressions that do not sound right and would be happy to incorporate those changes.